Biological age for chronic kidney disease patients using index model

Abu Bakar Shaiful Anuar 1 saab@um.edu.my
Syed Mohamed Shahruddin Sharifah Nazatul Shima 1
Ismail Noriszura 2
Wan Md Adnan Wan Ahmad Hafiz 3
1 Institute of Mathematical Sciences, Faculty of Science, Universiti Malaya , Kuala Lumpur , Malaysia
2 Department of Mathematical Sciences, Faculty of Science and Technology, Universiti Kebangsaan Malaysia , Selangor , Malaysia
3 Department of Medicine, Faculty of Medicine , Univesiti Malaya, Kuala Lumpur , Malaysia
Suner Aslı
Electronic publication date: 2022 Aug 1
Publication date: 2022
Volume: 10
Electronic Location ID: e13694
Received 2022 Jan 17; Accepted 2022 Jun 16
Copyright: © 2022 Abu Bakar et al.
Copyright year: 2022
Copyright holder: Abu Bakar et al.
License: This is an open access article distributed under the terms of the Creative Commons Attribution License, which permits unrestricted use, distribution, reproduction and adaptation in any medium and for any purpose provided that it is properly attributed. For attribution, the original author(s), title, publication source (PeerJ) and either DOI or URL of the article must be cited.
License URL: https://creativecommons.org/licenses/by/4.0/

Keywords: Ageing, Biological age, Chronic kidney disease, Illness severity

Funding: Universiti Teknologi MARA under Geran Penyelidikan Khas 600-RMC/GPK 5/3 (280/2020) This research was funded by Universiti Teknologi MARA under Geran Penyelidikan Khas (600-RMC/GPK 5/3 (280/2020)). The funders had no role in study design, data collection and analysis, decision to publish, or preparation of the manuscript.

==============================
The estimation of biological age (BA) is an important asymptomatic measure that can be used to understand the physical changes and the aging process of a living being. Factors that contribute towards profiling the human biological age can be diverse. Therefore, this study focuses on developing a BA model for patients with Chronic Kidney Disease (CKD). The procedure commences with the selection of significant biomarkers using a correlation test. Appropriate weighting is then assigned to each selected biomarker using the indexing method to produce a BA index. The BA index is matched to the age variation within the sample to acquire additional terms for the chronological age leading ultimately to the estimated BA. From a sample of 190 patients (133 trained data and 57 testing data) obtained from the University of Malaya Medical Centre (UMMC), Malaysia, the intensity of the BA is found to be between three to nine years from the chronological age. Visual observations further validate the high similarities between the training and testing data sets.

Introduction

The estimation of biological age (BA) is becoming increasingly popular as an asymptotic measure to understand changes in physical functionality as well as the ageing process of a living being. Over the chronological age (CA), which exhibits an exact figure signifying the period between birth and the present time of an individual, the BA is widely used to indicate the healthy and unhealthy ageing through variables that contribute to healthspan (Kim & Jazwinski, 2015). Furthermore, BA shows the health state of each individual and serves as a comparative measure between individuals of the same age and gender (Kang et al., 2017). Thus, it describes one lifetime behaviour informatively provided the common premise that physical functionality decline is parallel to deterioration of health condition and age increment.

Unhealthy individuals demonstrate only decreased function in CA, but functional BA is intended to represent different stages of ageing (Hayflick, 2007). Group of individuals that perceived ill-health exhibited higher biological ages compared to the healthier groups. In this respect, those with more significant functional biological age have a higher chance of death because they reached a more advanced ageing stage earlier than others. Thus, the traditional approach in measuring individual health according to CA is less appealing in today’s highly dynamic and rapidly changing global lifestyle.

Researchers have recently employed several statistical techniques to develop the BA model, primarily by incorporating multiple relevant biomarkers into the model. The BA model does not only predict ageing-related diseases but also considers the functional status during ageing (Jia et al., 2016). Several articles have been published on the measurement of BA using statistical methods. Multiple linear regression (MLR) remains one of the most widely used methods for calculating BA (Bae et al., 2008; Cho, Park & Lim, 2010; Jee, 2019; Jee & Park, 2017; Jia, Zhang & Chen, 2017; Levine, 2013; Nakamura & Miyao, 2007; Park et al., 2009; Krøll & Saxtrup, 2000). Nevertheless, MLR has been criticized for the multicollinearity risk besides the potential for estimates to regress toward the mean (Cho, Park & Lim, 2010). These suggest the MLR equation underestimates the individual BA in the older age while overestimating the younger age (Park et al., 2009). Principal component analysis (PCA) was proposed to overcome the disadvantage of MLR in the development of the BA formula (Kang et al., 2017; Cho, Park & Lim, 2010; Jee, 2019; Jia, Zhang & Chen, 2017; Levine, 2013; Nakamura & Miyao, 2007; Park et al., 2009). However, the PCA cannot avoid some of the statistical deficiencies of MLR (Klemera & Doubal, 2006). An alternative to this, the Klemera & Doubal Method (KDM) provides better precision in estimating BA than MLR and PCA methods (Cho, Park & Lim, 2010; Jee, 2019; Jia, Zhang & Chen, 2017; Levine, 2013). Although KDM gives the most reliable estimates in BA prediction, it involves complex calculations (Cho, Park & Lim, 2010).

This study focuses on developing the BA model for patients with Chronic Kidney Disease (CKD). The public-health effect from mortality due to this disease has not been fully assessed (Wen et al., 2008). Furthermore, CA does not give a good reflection on the time-dependent changes in kidney function (Rowland et al., 2018). To better understand an individual degree of ageing or life span and how CKD influences an individual degree of ageing, a new approach needs to be developed.

In this study, we develop the BA using the indexing method. An index number is the most common statistical method to measure changes in a set of data points besides summarizing and ranking a particular data set. Moreover, measuring BA by examining the index number keeps track of the original representation of the data and thus ensures the output resembles the empirical structure closely. During this indexing process, each selected biomarker is given a unique treatment corresponding to its severity level. Visual observations are also presented to justify the appropriateness of the method used.

Materials and Methods

IRB/Ethics approval

The data used in this study was approved by The Medical Research Ethics Committee, University of Malaya Medical Centre (MREC ID NO 2018428-6258). The committee granted permission to carry out the study within its facilities with common terms including; to adhere the instruction, guidelines and requirement by the committee. The Patient Information Sheet and Consent Form were waived by the committee. This retrospective study used the data based on the earlier initiated and completed studies in a new outlook.

Measure of correlation

The strength between two variables can be measured using Pearson’s correlation coefficient (Wackerly, Mendenhall & Scheaffer, 2014). A representative measure for this, the r-value, signifies both the magnitude and direction of the strength, that is, a closer value to ±1indicate high strength in the positive or negative direction. The r-value can be computed as: (1) r=n∑i=1n⁡xiyi−(∑i=1n⁡xi∑i=1n⁡yi)(n∑i=1n⁡xi2−(∑i=1n⁡xi)2)(n∑i=1n⁡yi2−(∑i=1n⁡yi)2)

where x and y are the value for the two variables and n is the total number of samples.

This study considers 10 biomarker relationships with the CA; height, weight, gender, BMI, creatinine, e-GFR, PB Systolic, BP diastolic, CTCA calcium score and CKD stage from CKD patients. BA biomarkers that have an absolute r-value greater than 0.15 were selected for inclusion in BA calculation (Jee, 2019; Park et al., 2009).

Weighted average method

This study proposes a weighted average method to estimate the weight for each significant biomarker. Note that the weight ranges from 0 to 1. Higher weight signifies a higher association between the health biomarkers to the BA index. The weight for biomarker i is computed as follows: (2) wi=|ri|∑j=1n⁡|rj|

where ri is the correlation coefficient of the ith biomarker computed using Eq. (1) and n is the total number of significant biomarkers. Note that the sum for all wi’s equals to one, ∑i=1nwi=1.

Indexing method

This study uses the indexing method for BA calculation. The index produced from this method represents the amount of change with respect to the base value. For each biomarker, the base value is set to be the normal value or the favourable health condition. Thus, each biomarker has a unique indexing assignment based on the medical measurement they carry. In brief, (3) Index(i)=|Measuredvaluei−NormalvalueiNormalvaluei|

where measured value is the current health reading of the patient while the normal value is the normal reading level for health biomarker i. Table 1 summarizes several common reading levels based on the standard clinical practice as well as work carried out in literature studies. The health biomarkers are categorized into several reading levels based on the severity of the postulated measurements.

Table 1 Reading level for body mass index, blood pressure, eGFR, and calcium score.

Health biomarkers	Reading level (Severity level)	Researchers	
Body Mass Index (BMI)
(in kg/m2)	less than 18.5
(at risk)	18.5 to 24.9
(normal)	25 to 29.9
(moderate risk)	greater than 30
(high risk)	(Fontana & Hu, 2014; Walpole et al., 2012)	
Blood Pressure (Systolic)
(in mmHg)	less than 120
(at risk)	120 to 130
(normal)	130 to 139
(moderate risk)	greater than 140 (high risk)	(Rahman, Chia & Yusoff, 2011)	
Blood Pressure (Diastolic)
(in mmHg)	less than 80
(at risk)	80 to 84 (normal)	85 to 89 (moderate risk)	greater than 90
(high risk)	(Rahman, Chia & Yusoff, 2011)	
eGFR
ml/min/1.73m2	≥90
Stage 1
(normal)	60 to 89
Stage 2
(low risk)	30 to 59
Stage 3
(moderate risk)	15 to 29
Stage 4
(moderately high risk)	<15 Stage 5
(high risk)	(Rastogi, Linden & Nissenson, 2008)	
CTCA Calcium Score	0
(no risk)	less than 100
(low risk)	101 to 400 (moderately high risk)	> 400
(high risk)	(Bhulani et al., 2013; Neves, Andrade & Monção, 2017)	

Index equations are then developed based on Table 1. Note that the normal reference value is taken as the mid-point of the normal reading level. The index equation for the BMI is given by: (4) IndexBMI={1−[413(x−18.5)],if18.5≤x≤21.75,1+[433(x−30)],if21.75<x≤30,1,otherwise.

The index equation for the systolic blood pressure is given by: (5) IndexSBP={1−[15(x−120)],if120≤x≤125,1+[115(x−140)],if125<x≤140,1,otherwise.

The index equation for the diastolic blood pressure is given by: (6) IndexDBP={1−[12(x−80)],if80≤x≤82,1+[18(x−90)],if82<x≤90,1otherwise.

The index equation for the eGFR is given by: (7) IndexeGFR={1,ifx≤15,90−x75,if15<x<90,0,ifx≥90.

The index equation for the CTCA is given by: (8) IndexCTCA={0,ifx≤100,x−100300,if100<x<400,1,ifx≥400.

Eqs. (4) to (8) formulate the framework for BA index calculation similar to medical practices for six biomarkers to detect the severity level.

Figures 1A to 1E show the aforementioned state graphically. Note that the health biomarkers index ranges from 0 to 1 where 0 indicates a favourable level of the health biomarker, index value from 0 to 1 indicates deteriorating health condition, while index value of 1 indicates a critically ill stage. The medical measures for each of the biomarkers are unique. Therefore, the index value can be a useful comparative tool across these measurements.

Figure 1 Biomarkers severity index.

In order to produce the biological index for each individual, each health biomarker index is multiplied by its corresponding weight. The weight proportionately signifies the contribution of each biomarker index to the BA index. Mathematically, the overall index for individual x is computed as follows: (9) IBioAgeforx=w1Index1,x+w2Index2,x+…+wnIndexn,x

where wi and Indexi,x are the weight and index for the ith health biomarker of individual x, respectively.

BA estimation

Several methods have been proposed to estimate the BA. Among these are the multiple linear regression (MLR) (Jia, Zhang & Chen, 2017) and principal component analysis (PCA) (Nakamura & Miyao, 2007). PCA method is derived from MLR and it reduces the effects of underestimated or overestimated BA (Jia, Zhang & Chen, 2017). Both methods show a linear relationship between BA and health parameters.

BA index models developed for predicting BA in this study also follow a linear relationship for individual general health status. It is developed based on the mathematical settings of the index method. The method suggests combining all individual subcomponents indices in one principal component (in this case, all health biomarkers into a single BA index). The subcomponent index measures the changes for each representative group of the biomarkers from the CA. Note, however, that the value of the index is not in year term. A common approach translating it into meaningful year-unit is by equation BAx=(IBioAgeforx×standarddeviation)+meanofCA. Because the sample data focuses on the individual kidney patients, the following adjustment was made to the BAx. (10) BAx=(IBioAgeforx×SD)+CAx

where BAx and CAx are the biological age and chronological age for individual x, respectively. The standard deviation SD is computed based on the chronological age of the sample.

Bland-Altman analysis

It is vital to observe the mean values to understand the nature of a prediction model. It follows that the degree of dispersion from the mean indicates the fitness of individual BA values (Jee & Park, 2017). Degrees of dispersion between BA and CA are commonly presented using the Bland-Altman plots. Bland-Altman plots the difference of CA and BA against the mean of the two measurements where it is easier to measure the magnitude, spot outlier, and see the data trend (Altman & Bland, 1983). If the differences are normally distributed, the mean differences should lie between d¯−1.96sandd¯+1.96s (95% confidence interval) where d¯ is the estimated mean difference and s is the standard deviation for the differences (Giavarina, 2015).

Results and discussion

Data

The study population consisted of 190 patients subject to stage 1 to stage 4 CKD. Patients were recruited from the inpatient clinic, University of Malaya Medical Center (UMMC), Kuala Lumpur, Malaysia. Chronological age varied from 35 to 82 years, and the study population included both males (115 patients) and females (75 patients). The age range was chosen to ensure that the population was old enough to be experiencing age-related changes in biomarkers.

Physical measurements include gender, height, weight, body mass index (BMI), systolic blood pressure (SBP), diastolic blood pressure (DBP), CTCA Calcium Score (CTCA), creatinine, CKD stage and eGFR. It is observed that the CKD stage for the data involved was between 2 and 5. More specifically, 4% were in stage 2, 50% were in stage 3, 38% were in stage 4 and 8% were in stage 5. Table 2 shows the mean result, standard deviation (SD) and data range of CKD patients. For validation purposes of the BA model, 70% of the data were used as the training set (i.e., 133 data) while the remaining 30% were used for the testing set (i.e., 57 data).

Table 2 General characteristics of the CKD patient biomarkers.

Parameters	Male
(n = 115)	Female
(n = 75)	Combined
(n = 190)	
Age (years)	62.88 ± 9.463	61.97 ± 9.988	62.52 ± 9.657	
Weight (kg)	76.77 ± 12.305	68.60 ± 14.347	73.54 ± 13.709	
Height (m)	1.66 ± 0.0647	1.523 ± 0.060	1.604 ± 0.091	
BMI (kg/m2)	27.93 ± 0.392	29.469 ± 5.109	28.535 ± 4.6328	
CTCA calcium score	558.997 ± 733.855	291.853 ± 470.753	453.546 ± 654.786	
Creatinine (µmol/L)	242.11 ± 130.025	229.59 ± 105.253	237.17 ± 120.718	
eGFR (mL/min/1.73m2)	35.513 ± 14.857	28.481 ± 12.432	32.737 ± 14.336	
BPSystolic (mmHg)	146.88 ± 23.808	148.20 ± 21.075	147.4 ± 22.720	
BPDiastolic (mmHg)	76.47 ± 14.906	76.23 ± 13.736	76.37 ± 14.419	
CKD Stage	3.41 ± 0.687	3.67 ± 0.081	3.51 ± 0.703	
Note:

Mean ± SD.

Findings

The development of BA using the indexing method involved several sequences; correlation analysis, computation of weighted average for selected health biomarkers, construction of BA indices from the index equations and the estimation of BA based on sample variations.

Biomarkers that show an absolute correlation coefficient value exceeding 0.15 were selected for BA estimation. An associated significance test that examined the correlation between the biomarkers and the chronological age for the training data set is summarized in Table 3. Seven biomarkers were found significant for inclusion in the BA estimation. It was observed that creatinine, weight, BMI, eGFR, BP systolic and BP diastolic decreased as age increased, while only the CTCA Calcium increased with age. Height, CKD stage and gender had a low correlation with age and were thus excluded from the BA estimation.

Table 3 p-values for the significance test.

Parameters	p-values	
Weight (kg)	0.003	
Height (m)	0.056	
BMI (kg/m2)	0.023	
CTCA calcium score	0.002	
Creatinine (µmol/L)	0.010	
eGFR (mL/min)	0.004	
BPSystolic (mmHg)	0.041	
BPDiastolic (mmHg)	0.000	
CKD stage	0.081	
GenderT	0.421	
Note:

Trained data, n = 133.

Note that some biomarker features had almost similar clinical implications and expressed high inter-correlation. It indicates the existence of redundancy. To cater for this redundancy, one biomarker with more substantial significance was selected. It is observed from Table 4 that weight and BMI had an absolute correlation coefficient of more than 0.15 (−0.234 and −0.173, respectively) and both showed high correlation with each other (0.787). Due to the marginal difference in their correlation with age and clinical significance of the BMI, it was selected for BA estimation. Furthermore, BMI was more reliable to indicate the individual weight, either normal, overweight or obese (North American Association for the Study of Obesity, 2000); a similar procedure was used for selection between the creatinine and eGFR. Both biomarkers represent the renal function. Creatinine and eGFR showed mild inter-correlation (0.641) in addition to the absolute correlation coefficient of more than 0.15 (−0.202 and −0.233, respectively). Therefore, eGFR was selected for the BA estimation. With respect to its significance, measurement of eGFR is the most reliable assessment of renal function in CKD (Bostom, Kronenberg & Ritz, 2002) where it is used as an index of renal function in clinical practice (Perrone, Madias & Levey, 1992).

Table 4 Correlation coefficients (Pearson) between CA and biomarkers.

Parameter	Age	CTCA calcium score	Creatinine	Weight	Height	BMI	eGFR	BPSystolic	BPDiastolic	CKD stage	Gender	
Age	1.000	0.249	−0.202	−0.234	−0.138	−0.173	−0.233	−0.151	−0.411	0.122	0.017	
	1.000	0.021	0.045	0.106	0.007	−0.056	0.009	−0.134	0.062	0.226	
		1.000	0.042	0.113	−0.018	−0.641	0.179	0.021	0.716	0.066	
			1.000	0.515	0.787	0.544	0.012	0.095	−0.452	0.369	
				1.000	−0.108	0.292	0.087	0.115	−0.287	0.713	
					1.000	0.399	−0.045	0.023	−0.306	−0.081	
						1.000	−0.083	0.203	−0.862	0.276	
							1.000	0.542	0.121	−0.016	
								1.000	−0.138	0.053	
									1.000	−0.259	
										1.000	

The weight for each parameter was derived based on the correlation analysis. The higher the correlation, the higher the weighted for the BA parameters. As shown in Table 5, BP Diastolic, CTCA and eGFR were the three highest contributors to the index value for predicting the BA.

Table 5 Average weighted for significant BA biomarkers.

Parameters	Weightage	
Weight	0.142	
BMI	0.105	
CTCA	0.151	
Creatinine	0.122	
eGFR (mL/min)	0.141	
BP Systolic (mmHg)	0.091	
BPDiastolic (mmHg)	0.249	

Accordingly, five biomarkers, including, BMI, CTCA calcium, eGFR, BP systolic and BP diastolic were selected to estimate the BA index. The weightage for each of the selected biomarkers was then computed using Eq. (2) to arrive at the following BA index: (11) IBioAge=0.1422IBMI+0.2046ICTCA+0.1915IeGFR+0.1241IBPSystolic+0.3377IBPDiastolic

Table 6 summarizes the estimated BA based on the BA index of Eq. (11). It is evident that all estimated BAs were higher than their corresponding CAs for CKD patients. Note, however, that the increase in BA varied for patients with identical CKD stages, acknowledging other competing factors that compensate for the difference. It was observed that the gain in BA for the 133 training data set ranges from 3 to 9 years with a mean of 7 years.

Table 6 BA estimated using the BA index.

No.	CA	CKD stage	BA index	BA	No.	CA	CKD stage	BA index	BA	
1	64	4	0.5240	69	68	70	3	0.8971	78	
2	60	4	0.6699	66	69	56	3	0.9104	65	
3	48	3	0.5611	53	70	64	5	0.9575	73	
4	65	4	0.5914	71	71	65	3	0.9392	74	
5	67	4	0.7351	74	72	37	3	0.6988	44	
6	63	5	0.9507	72	73	53	3	0.6813	59	
7	60	5	0.7954	67	74	81	3	0.8380	89	
8	59	3	0.7737	66	75	36	2	0.6562	42	
9	70	4	0.9129	79	76	73	4	0.8726	81	
10	66	3	0.9264	75	77	68	3	0.6789	74	
11	61	3	0.7143	68	78	70	4	0.8573	78	
12	82	4	0.7948	89	79	74	3	0.6681	80	
13	57	5	0.8876	65	80	48	4	0.9191	57	
14	48	2	0.6292	54	81	64	3	0.8854	72	
15	67	4	0.6962	74	82	61	4	0.7756	68	
16	43	3	0.8947	51	83	37	4	0.6919	44	
17	81	5	1.0000	90	84	65	4	0.7499	72	
18	55	3	0.3852	59	85	78	3	0.7795	85	
19	63	3	0.7392	70	86	77	4	0.9145	86	
20	63	3	0.7570	70	87	69	4	0.8449	77	
21	68	4	0.9649	77	88	58	3	0.7076	65	
22	60	4	0.7578	67	89	63	3	0.7245	70	
23	67	4	0.7214	74	90	63	3	0.9308	72	
24	58	5	0.9680	67	91	70	3	0.7933	77	
25	58	4	0.7490	65	92	74	4	0.8170	82	
26	56	4	0.8093	64	93	53	4	0.6986	60	
27	43	4	0.9002	51	94	67	4	0.7660	74	
28	62	4	0.8951	70	95	71	3	0.7973	79	
29	75	3	0.8297	83	96	53	5	0.6726	59	
30	55	4	0.9661	64	97	52	3	0.6456	58	
31	62	4	0.6336	68	98	60	3	0.8970	68	
32	63	4	0.7361	70	99	73	3	0.7774	80	
33	72	4	0.7405	79	100	61	5	0.5983	67	
34	76	4	0.8005	84	101	66	3	0.5996	72	
35	43	4	0.6770	49	102	71	4	0.9006	79	
36	59	3	0.9323	68	103	78	3	0.6414	84	
37	72	3	0.6619	78	104	64	4	0.6862	70	
38	64	4	0.6324	70	105	50	3	0.6637	56	
39	72	4	0.8376	80	106	68	3	0.7547	75	
40	46	3	0.7040	53	107	53	3	0.8595	61	
41	75	4	0.7902	82	108	75	3	0.7372	82	
42	69	3	0.6955	76	109	63	3	0.9176	72	
43	66	4	0.6984	73	110	58	4	0.6760	64	
44	68	2	0.4181	72	111	82	4	0.9827	91	
45	64	3	0.6099	70	112	68	3	0.7254	75	
46	55	4	0.9653	64	113	67	3	0.7050	74	
47	63	4	0.7009	70	114	61	5	0.8571	69	
48	63	3	0.3643	66	115	51	4	0.7512	58	
49	69	4	0.8109	77	116	54	5	0.7234	61	
50	69	3	0.8070	77	117	72	3	0.5938	78	
51	76	4	0.6379	82	118	67	3	0.7171	74	
52	68	4	0.7109	75	119	77	4	0.8781	85	
53	65	3	0.5285	70	120	60	3	0.6174	66	
54	65	3	0.8299	73	121	64	3	0.7517	71	
55	59	3	0.7763	66	122	62	3	0.5667	67	
56	69	4	0.8914	77	123	82	4	0.9089	91	
57	76	3	0.9309	85	124	74	4	0.8258	82	
58	69	4	0.7874	76	125	74	4	0.8620	82	
59	63	5	0.6161	69	126	62	3	0.7566	69	
60	48	3	0.6912	55	127	63	4	0.7086	70	
61	59	5	0.9720	68	128	67	3	0.7509	74	
62	53	2	0.4258	57	129	59	3	0.6114	65	
63	56	2	0.6054	62	130	71	5	0.9078	80	
64	65	2	0.7176	72	131	71	4	0.6842	77	
65	69	4	0.9307	78	132	65	3	0.6418	71	
66	58	3	0.6751	64	133	71	3	0.9161	80	
67	57	3	0.6600	63						

Overall, the BA for patients suffering kidney disease increased by 5% to 16% from its CA. Figure 2 shows that about 65% of kidney patients in stage 3 and stage 4 increased 9% to 12% from their CA. It indicated that on average the CKD patients at these stages gain between 5 to 9 years from their CA biologically.

Figure 2 BA increment from CA by CKD stage.

The Bland-Altman plot in Fig. 3 exhibits the differences in CA and BA against the mean with a 95% confidence interval. All plots are shown below the zero value because this study utilizes data for patients diagnosed with CKD. Note however, the plot does not indicate whether the limits are acceptable or not. The judgment is based on clinical necessity. It is observed that most of the plots lie between the 95% confidence interval (CI), that is, they are inside the limit of agreement (LOA).

Figure 3 Bland-Altman plot.

To explain further, the differences between CA and BA are plotted against the z-score of a standard normal distribution in Fig. 4. It was observed that the majority of the plots fall along the straight line, which suggests the difference between CA and BA to follow the normal distribution. Furthermore, the correlation was found to be 0.9715 and the bell test using kurtosis test (0.170) indicates the degree of tailedness in the frequency was close to perfect normal distribution. In addition, the skewness test gives a value of 0.395 where the test value was between −0.5 and 0.5 or nearly zero, which justifies the assumption of normal distribution.

Figure 4 Scatter plot for differences between CA and BA vs. the Z-score.

To further validate the BA index model, the testing dataset with 57 CKD patients was examined. Figure 5 shows the proximity between BA for the testing dataset and BA for the training dataset. The BA of CKD patients increases as the CA increase due to the severity of the biomarkers measurements. On average, the age for patients diagnosed with CKD (stage 1 to stage 4) increases by 7 years from their CA (64 years old) to BA (71 years old). Nevertheless, we conceived the small size of the sample and its representation of the population as the limitation of the study.

Figure 5 CA vs. BA.

Conclusions

Recent studies have shown the importance of assessing the functional biomarkers for predicting ageing-related diseases. Ageing-related disease influences the BA of a person. Several statistical approaches have been observed in constructing the BA model, namely, the MLR, the PCA and the KDM (Levine, 2013). Each method owns its deficiency in the measurement of BA. Therefore, the indexing method is proposed to address issues, especially with redundant biomarkers (through manual examination of redundant biomarkers and selection of the relevant biomarker by experience), over or underestimated BA for a particular age group (by concentration to a particular disease group), and complicated calculations (through a tractable computation for each biomarker).

The estimation of BA using the indexing method proposed in this study facilitates a tractable form for each health biomarker. The initially calculated BA index represents the illness severity level for the CKD patients which contributes to proportionate gain in the BA. The level of severity is categorized into high risk, normal risk and at risk so that each patient may be assessed individually. Ten biomarkers were examined to see their appropriateness for inclusion in the BA estimation.

The results of this study show that patients with CKD between stage 2 to stage 5 experience gain in BA between 3 to 9 years. This finding may serve the medical practitioners a precaution for treatment in addition to current measurement facilities as it provides a comprehensive reading of the manifold biomarkers. The result is further validated with trained data and visual observations. Notwithstanding, increasing the sample size for the study and inclusion of diverse population may enhance the reliability of the end result.

Besides its practicality in the medical field, BA can be potentially useful in many areas, including the insurance industry where age and health play a central role in the premium calculation. Mortality projection based on the BA can be an exciting exploration of human lifetime behaviour that incorporates health biomarkers. In addition, an interesting future work is to develop methods which enables a fair comparison between the various biological age approaches.

Supplemental Information

Supplemental Information 1 The raw measurements of quantities for patients with chronic kidney disease.

Click here for additional data file.

The authors acknowledge the University of Malaya Medical Center (UMMC) for providing data for this research. The authors also thank the Editor and the Reviewers for their thoughtful comments and constructive suggestions.

Additional Information and Declarations

Competing Interests

Author Contributions

Human Ethics

Data Availability

The authors declare that they have no competing interests.

Shaiful Anuar Abu Bakar conceived and designed the experiments, performed the experiments, authored or reviewed drafts of the article, and approved the final draft.

Sharifah Nazatul Shima Syed Mohamed Shahruddin conceived and designed the experiments, performed the experiments, prepared figures and/or tables, authored or reviewed drafts of the article, and approved the final draft.

Noriszura Ismail performed the experiments, analyzed the data, prepared figures and/or tables, authored or reviewed drafts of the article, and approved the final draft.

Wan Ahmad Hafiz Wan Md Adnan performed the experiments, analyzed the data, prepared figures and/or tables, authored or reviewed drafts of the article, and approved the final draft.

The following information was supplied relating to ethical approvals (i.e., approving body and any reference numbers):

The Medical Research Ethics Committee, University of Malaya Medical Centre granted approval to carry out the study within its facilities (MREC ID NO 2018428-6258).

The following information was supplied regarding data availability:

The raw data is available in the Supplemental File.

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
