# Peer review of "Biological age for chronic kidney disease patients using index model"

_PeerJ, doi:10.7717/peerj.13694_

## Round 0.1 · original submission · Major Revisions

Your manuscript has been reviewed and requires several modifications prior to making a decision. The comments of the reviewers are included at the bottom of this letter. Reviewers indicated that the methods section should be improved. I agree with the evaluation and I would, therefore, request for the manuscript to be revised accordingly.

Reviewer 1 ·

Basic reporting

The manuscript addresses potentially important and useful issues in the biological age estimation.
Thus, from my opinion the manuscript can be a good contribution to the field, the style and the length are appropriate for full development of the ideas, and I therefore recommend its acceptance for publication after a minor revision.

Experimental design

Line 27: What do you mean by “intensity”? Please clarify the idea.

Lines 95-104: Correlation analysis is restricted to Pearson's correlation coefficient for quantitative variables. However, taking into account the use of categorical variables (such as gender or CKD stage), were the results obtained with other coefficients (such as Spearman's or Kendall's) analysed?

Lines 156-159: Please, explain formula (10), namely which standard deviation is used and whether BA_x and CA are from the same individual.

Line 192: Please, identify the applied statistical test.

Validity of the findings

Taking into account the formula used to compute BA estimates, does the BA index allow to estimate the biological age or only the distance between the chronological age and the biological age? That is, to assess how greater is the biological age than the chronological age?

Additional comments

Some indices are missing from formula (1)

Line 143: “I_i” should be replaced by “Index_i”

Line 166-167: Please, define \bar{d} and s.

Line 168: Please delete "Add your materials and methods here."

Lines 342-344: Improve reference 26

Reviewer 2 ·

Basic reporting

The paper is somewhat clear, but some important details are missing or unclear. I suggest the authors provide sufficient background for a large audience.

Experimental design

In this study, the authors develop the BA using the indexing method. The results of this study show that patients with CKD between stage 2 to stage 5 experience gain in BA between 3 to 9 years.

1. I recommend that authors use three-line tables for all tables (no vertical lines, only top, bottom, and column lines).
2. Please indicate how the method used in this article differs from MLR and PCA. Does this method avoid the deficiencies of MLR, PCA, and KDM?
3. Please add figure legend, a simple explanation of each picture makes it easier for readers to understand.
4. I suggest authors use different types of statistical approaches to acquire the BA prediction models, then comparisons of biological ages calculated by the KDM, PCA, and MLR approaches to corresponding chronological ages.

Validity of the findings

The paper contributes some new ideas. But the validity of the findings in this study is somewhat limited by the small sample sizes.

---

## Round 0.2 · Minor Revisions

Your manuscript has been reviewed and still requires modifications prior to making a decision. The comments of the reviewer are included at the bottom of this letter.

Reviewer 1 ·

Basic reporting

The authors have taken into account the reviewers’ comments in the revision of the manuscript.
Therefore, I recommend its acceptance for publication.
Nonetheless, the English writing should be carefully revised throughout all the text.
Please also confirm the numbering of the bibliographic references, namely those in Table 1. As it stands, there are no citations to references 24 to 28.

Experimental design

No comment.

Validity of the findings

No comment.

Additional comments

Lines 111-112 - In the pdf file it is shown "Error! Reference source not found.".
Equation (9) and line 142 - Are the weights equal for all individuals? If so, they do not need the index x.
Line 156 - "BioAge" should be replaced by "BioAge for x" in the index of I
Line 199 - "biomarkers feature almost similar" should be replaced by "biomarkers feature have almost similar".
Line 238 - "to follow the normal distribution Z ~ N(0,1)." should be replaced by "to follow a normal distribution.". The difference between CA and BA do not have zero mean neither unitary standard deviation as stated: Z ~ N(0,1).
Line 278 - "enables a just comparison" should be replaced by "enables a fair comparison".

Reviewer 2 ·

Basic reporting

N/A

Experimental design

N/A

Validity of the findings

N/A

Additional comments

N/A

---

## Round 0.3 · accepted · Accept

Dear Dr. Abu Bakar and colleagues:

Thanks for revising your manuscript based on the concerns. I now believe that your manuscript is suitable for publication. Congratulations! I look forward to seeing this work in print. Thanks again for choosing PeerJ to publish.

Best,
Aslı